# Polycaprolactone/Polyethylene Glycol Blended with *Dipsacus asper* Wall Extract Nanofibers Promote Osteogenic Differentiation of Periodontal Ligament Stem Cells

**DOI:** 10.3390/polym13142245

**Published:** 2021-07-08

**Authors:** Te-Yang Huang, Mohsen Shahrousvand, Yu-Teng Hsu, Wen-Ta Su

**Affiliations:** 1Department of Orthopedic Surgery, Mackay Memorial Hospital, Taipei 104217, Taiwan; haungt33@gmail.com; 2Department of Caspian Faculty of Engineering, University of Tehran, Tehran 1417935840, Iran; tina422942@gmail.com; 3Department of Chemical Engineering and Biotechnology, National Taipei University of Technology, Taipei 106344, Taiwan; cf1811@ntut.edu.tw

**Keywords:** *Dipsacus asper* wall, periodontal ligament stem cell, osteogenesis, electrospinning, PI3K

## Abstract

*Dipsacus asper* wall (DA) is an ancient Chinese medicinal material that has long been used to maintain the health of human bones. The present study aimed to evaluate the osteogenic differentiation of periodontal ligament stem cells (PDLSCs) of *Dipsacus asper* wall extracts (DAE). Microwave-assisted alcohol extraction of 100 mesh DA powder under optimal conditions can obtain 58.66% (*w/w*) yield of the crude extract. PDLSCs have excellent differentiation potential. PDLSCs treated with DA extract (DAE) underwent osteogenesis, exhibiting a higher expression of the Col-1, ALP, Runx2, and OCN genes, and had a 1.4-fold increase in mineralization, demonstrating the potential of DAE to promote osteogenic differentiation. After the addition of PI3K inhibitor LY294002, the expression of osteogenic genes was significantly inhibited, confirming that PI3K is an important pathway for DAE to induce osteogenesis. Mix DAE with polycaprolactone/polyethylene glycol (PCL/PEO) to obtain nanofibers with a diameter of 488 nm under optimal electrospinning conditions. The physical property analysis of nanofibers with and without DAE includes FTIR, mechanical strength, biodegradability, swelling ratio and porosity, and cell compatibility. When cells induced by nanofibers with or without DAE, the mineralization of PDLSCs cultured on PCL/PEO/DAE was 2.6-fold higher than that of PCL/PEO. The results of the study confirm that both DAE and PCL/PEO nanofibers have the effect of promoting osteogenic differentiation. In order to obtain the best induction effect, the optimal amount of DAE can be discussed in future research.

## 1. Introduction

*Dipsacus asper* wall (DA) is a Chinese traditional medicinal material composed mainly of polysaccharides, cyclic olefin ether glycosides, oleanolic acid, triterpenoid saponins, volatile oils, and β-sitosterol [1]. In a previous study, Sun et al. purified the total polysaccharides from DA and fed these continuously to ovariectomized female rats for 12 weeks, ultimately confirming that the total polysaccharides in the DA inhibited osteoporosis [2]. Chen et al. purified the water-soluble polysaccharides from DA and confirmed that these could inhibit the growth of human osteosarcoma cells by regulating the expression of the PI3K protein [3]. Niu et al. induced MC3T3-E1 osteoblasts with asperosaponin VI (a saponin component from DA), which promoted the expression of osteogenesis-related genes and proteins; they concluded that the osteogenic promotion occurred through the P38 and ERK1/2 pathways [4]. The purified total saponins from DA were also tested in MC3T3-E1 cells in vitro, and the total saponins in DA were confirmed to promote osteogenesis through the BMP-2/MAPK/Smad-dependent Runx2 pathway [5]. Alcohol extraction of DA can get a mixture, which mainly contains polysaccharides and triterpene saponins, and the extract of DA has a potential effect on the induction of osteogenic differentiation of stem cells.

Polycaprolactone (PCL) is a polyester polymer with strong hydrophobicity, high mechanical strength, and complete biodegradability. PCL certified by the Food and Drug Administration (FDA) to be biocompatible, and PCL can therefore be applied to structures that will be implanted within the human body. Due to the strong hydrophobicity of this polymer, cell adhesion is insufficient, so it is usually necessary to add other materials to improve cell adhesion. Therefore, Remya et al. improved the hydrophilicity and biodegradability of PCL by adding polyethylene glycol [6]. Polyethylene oxide (PEO) has extremely high hydrophilicity, is miscible with water in any proportion, is soluble in most common solvents, it is biologically inert, and usually does not cause allergic reactions. PEO is often used as a drug solid or a film coating to control the rate of drug release. In regenerative medicine, PCL blends with PEO is a good biomedical material, suitable for drug release. There are many biorenewable resources that have been applied to nanocomposites in regeneration medicine [7].

Tissue engineering involves cells, scaffold, and signals to which the principles of life science and engineering are applied in order to develop biological substitutes that can be implanted into the human body to restore, maintain, or improve tissue function [8,9,10,11]. Porous three-dimensional (3D) scaffolds, such as nanofibers fabricated by electrospinning, have greater advantages in repairing large bone defects [12]. The principle of electrospinning is the application of a high voltage, of thousands or even tens of thousands of volts, to a needle loaded with a polymer solution. When the static electricity in the solution increases, the potential energy of the charge exceeds the surface tension, the solution will eject, and the swinging of the solution occurs entirely inside the electric field. During the process, the solvent will volatilize, resulting in the even distribution of the filamentous polymers in a solid state on the collection plate. Electrospinning can form nanofibers that are much smaller than the size of cells and can also form porous structures, which are good for stimulating the growth of cells through environmental cues within the body. In 2002, Li et al. proposed the advantages of electrostatic nanofiber technology for use at the level of tissue engineering [13], Yoshimoto et al. further confirmed the great potential of electrostatic nanofibers as bone tissue engineering scaffolds [14], Alven and Aderibigbe reviewed the PVA/PCL hybrid nanofiber for wound dressing application [15].

Stem cells can produce undifferentiated cells after division and also exhibit the potential to differentiate and become specialized cells [13]. Mesenchymal stem cells (MSCs) are a type of pluripotent stem cell that express CD166, CD105, CD73, and CD90, and do not express CD45, CD34, CD14, CD11b, CD79a, CD19, or HLA-DR. MSCs can differentiate into various germ layer tissues [16,17] and are advantageous as they have great differentiation potential, strong proliferation ability, low immunogenicity, convenient collection procedures, and no restrictions on use due to ethical issues. Periodontal ligament stem cells (PDLSCs) are a type of MSCs. They were first discovered in the periodontal ligament by Seo et al. in 2004 [18] and were successfully isolated and purified by Gay et al. in 2007 [19]. PDLSCs have excellent differentiation potential, and their ability to differentiate into hard bone, cartilage, and fat cells is comparable to that of bone marrow mesenchymal stem cells (BMSCs) [20]. In 2010, Iwata et al. improved the technology for the separation of PDLSCs, enabling extraction with up to 96.7% purity of the target cells, and showed that PDLSCs have better osteogenic ability than BMSCs under certain conditions [21]. Surface topography and nanostructures of biomaterials play an important role in cellular attachment, migration, and also influence stem cell proliferation and differentiation. Nanofibers composed of ultrafine and continuous fibers possess high porosity, variable pore-size distribution, and high surface-to-volume ratios, as well as resembling a natural ECM topography, which have recently attracted attention in the field of tissue engineering and regenerative medicine [22].

Osteogenic differentiation refers to the process through which MSCs or osteoprogenitor cells differentiate into mature osteocytes [23]. After stem cells proliferate, they begin to differentiate into osteoprogenitor cells, which subsequently differentiate into osteoblasts. Osteoblasts secrete bone matrix and deposit minerals. During their final stage, osteoblasts are either apoptotic or become incorporated into bone matrix as osteocytes [24]. Cells have various functions at different stages, and therefore exhibit differences in gene and protein expression levels. Observation of the expression of these specific markers is helpful to understand the progress and effects of cell differentiation.

Microwave-assisted extract is an alternative technique converts electromagnetic energy into reactants, which can achieve higher yields, selectivity, and reaction rates while using less solvent, consuming less energy, and producing less waste than traditional extraction. The present study aims to confirm the effect of DA extract (DAE) in promoting osteoblastic differentiation of PDLSCs. The DAE was blended with PCL/PEO solution to form 3D nanofibers. Cellular viability was measured via MTT or CCK-8 assay. Real-time polymerase chain reaction (RT-PCR), western blotting, and Alizarin Red S staining were performed to determine matrix maturation and mineralization.

## 2. Materials and Methods

### 2.1. Isolation and Identification of PDLSCs 

Healthy molars were extracted from donors following a standard operation procedure approved by the Institutional Review Board (IRB) of the Dental Clinic of Kaohsiung Medical University (Taiwan) (KMU-26434). Periodontal ligament (PDL) cells were separated from third molars using enzymatic digestion. PDL tissues were digested in a solution of 3 mg/mL collagenase type I (Worthington Biochemical, Lakewood, NJ, USA) and 4 mg/mL dispase (Sigma-Aldrich, St. Louis, MO, USA) for 1 h at 37 °C. Single-cell suspensions were obtained by passing digested samples through a 70 μm strainer (Falcon; Thermo Fisher Scientific, Waltham, MA, USA). Cell suspensions were centrifuged at 1000 rpm for 10 min, and single cells were resuspended in culture medium composed of α-MEM (Gibco; Thermo Fisher Scientific) with 10% fetal bovine serum (FBS), 100 μM L-ascorbic acid 2-phosphate (Sigma-Aldrich), 1% antibiotic-antimycotic (Gibco), and then incubated at 37 °C with 5% CO_2_. Subculturing was performed at 80% confluence for ordinary cultures, and the medium was changed every two days. The identification of isolated PDLSCs was assayed the cell surface molecules including CD34, CD45, CD73, CD90, CD105, CD146, and CD166 by FACSCalibur flow cytometer (BD Bioscience, Bergen, NJ, USA).

### 2.2. Preparation the Extracts of DA

DA was purchased from a local grass shop in Taiwan, and then the samples were crushed into a powder and sieved through a 100-mesh screen. Different parameters were adjusted (temperature, extraction time, solid/solvent (g/mL) ratio and microwave power) to optimize the process conditions to provide the maximum yield of extraction in microwave-assisted extraction (MAE). DA powder, in 5.0 g samples, was mixed with 70% ethanol and extracted using a microwave-assisted machine (MAS-II Plus, Sineo, Shanghai, China) to determine the maximum yield of the ethanol extract.

### 2.3. Fabrication of PCL/PEO Nanofibers Containing DAE

To a brown bottle, 0.2 g of PEO (Mw: 300,000 g/mol) and 0.8 g of PCL (Mw: 80,000 g/mol) were added, followed by 1 mL of DMSO with or without 0.1 g DAE, 1 mL of N,N-dimethylformamide (DMF), and 8 mL of chloroform (CF). The solution was then stirred for at least one day to ensure sufficient mixing. The homogeneous solution of PCL/PEO, with or without DAE, was loaded into a 10-mL syringe connected to a 23-gauge needle with a distance of 178 mm between the tip and the aluminum foil collector, and the device was placed into the syringe pump of an electrospinning system. The solution flow rate was fixed at 1.0 mL/hr. The solute ration, solvent ration, and electric voltage will be adjusted to the best condition of electrospinning. Electrospinning was performed for 10 h to obtain a nanofibrous membrane, and then the membrane was removed from the aluminum foil and air-dried for 24 h.

### 2.4. Characterization of Nanofiber

To analyze the appearance and diameters of the PCL/PEO and PCL/PEO/DAE nanofibers under a scanning electron microscope (SEM), the fibers were dehydrated in ethanol solutions, sequentially increasing to dehydration in 100% ethanol. The specimens were coated with a thin layer of gold (Cressington 108 Manual; Ted Pella Inc., Redding, CA, USA) and then examined under a SEM (FEI Quanta 200; Thermo Fisher Scientific). The diameter of the fabricated fibers was quantified using image analysis software (Image J, 1.43S, National Institutes of Health, Bethesda, MA, USA), and the fiber average diameter was determined by measuring the diameters of 100 randomly selected fibers. Composition analysis was performed using Fourier-transform infrared spectroscopy (FTIR; FT-720; HORIBA, Kyoto, Japan).

The swelling ratio of the nanofiber was assayed and calculated using the following Equation (1): Swelling ratio (%) = [(Ws − Wd)/Wd] × 100%(1)
where Wd is the fiber’s dry weight and Ws is the weight of the swollen fiber.

The porosity of the PCL/PEO nanofiber was measured using the liquid displacement method and calculated using the following equation: Porosity (%) = (V_i_ − V_w_)/(V_t_ − V_w_) × 100%(2)
where V_i_ is the initial known volume of water, V_t_ is the total volume of water and water-impregnated fibers, and V_w_ is the residual water volume removed by the fiber.

The experiments were performed five times, and the average porosity was determined. Mechanical strength of the PCL/PEO and PCL/PEO/DAE nanofibers was performed using a universal testing machine (PT-003; Kotsao, Taipei, Taiwan). The tested sample was cut into a thin strip, and a tensile test was conducted at a crosshead speed of 50 mm/min at room temperature. The biodegradability of the PCL/PEO and PCL/PEO/DAE nanofibers was assessed by soaking them in phosphate buffer solution (PBS) at 37 °C for 20 days and then obtaining the residual weight of the fibers to calculate the degradation rate.

### 2.5. Cellular Viability vs. DAE Treatment

To evaluate the cellular growth performance of PDLSCs treated with DAE, cells from passages 3 to 5 were seeded into 96-well plates at 1 × 10^4^ cells/well in the cell culture medium described above, which was replaced every three days for seven days. This is a colorimetric assay was based on 3-(4,5-dimethylthiazol-2-yl)-2,5-diphenyltetrazolium bromide (MTT, Sigma-Aldrich) for assessing cell metabolic activity as an indicator for cell viability, proliferation, and cytotoxicity. The well containing the proliferated cells was treated with 5 mg/mL MTT at 37 °C for 4 h. The cultivated medium was removed, and formazan was solubilized in DMSO. The metabolized MTT was measured based on optical density at 570 nm using a spectrophotometer (Multiskan FC, Thermo Fisher Scientific). The nanofiber is clamped flat with a microcentrifuge tube and cut to form a culture well, then sterilized with autoclave. PDLSCs from passages 3 to 5 were seeded into the nanofiber at 1×10^4^ cells/well. The cellular viability of PDLSCs on nanofibers was measured using the Cell Counting Kit-8 (CCK-8; Dojindo, Kumamoto, Japan) for 21 days. The proliferated cells on nanofibers were treated with 10% CCK-8 reagent at 37 °C for 4 h, and then optical density at 450 nm was measured using a spectrophotometer (Multiskan FC, Thermo Fisher Scientific). 

### 2.6. Calcium Quantification

Extracellular matrix calcium deposition to form bone nodules occurs in the later stage of osteogenic differentiation; therefore, the calcification of cells on days 7, 14, and 21 were measured to assess Ca quantification. The accumulated Ca in the secreted mineral matrix of the osteoblasts was quantified using Alizarin Red S staining. All samples were washed twice with PBS then immersed into 95% ethanol solution for 30 min. The samples were then stained with 1% Alizarin Red S for 10 min, washed thrice with PBS, and finally solubilized with 10% cetylpyridinium chloride. Images of stained cells were captured using an optical microscope (CKX53; Olympus, Tokyo, Japan), the amount of deposited calcium in the stained cells was calculated using standard solutions and measuring the absorbance at 570 nm (Multiskan FC, Thermo Fisher Scientific).

### 2.7. Reverse Transcription-Polymerase Chain Reaction (RT-PCR)

The total RNA from differentiated PDLSCs was extracted using TRIzol reagent (Ambion®, Life Technologies™, Carlsbad, CA, USA) for 10 min. RNA quantity was determined using a NanoDrop 2000 spectrophotometer (Thermo Fisher Scientific). cDNA was synthesized from 1000 ng of RNA using the Super Script® III Reverse Transcriptase kit (Bio-Rad, Hercules, CA, USA) with a thermocycler (5 min at 25 °C, 20 min at 46 °C, 1 min at 95 °C). Real-time polymerase chain reaction (RT-PCR) was performed using Smart Quant Green Master Mix (Protech Bio, Taipei, Taiwan) according to the manufacturer’s protocol. An initial denaturation was performed at 95 °C for 15 min followed by 60 cycles of 95 °C for 15 s and 60 °C for 60 s. The relative gene-expression fold change was determined using the 2^−∆∆Ct^ method and normalized to transcripts of the housekeeper gene GAPDH. The PCR primers were as follows: Col-1 (Forward: 5′-TGC TTG AAT GTG CTG ATG ACA GGG-3′, Reverse: 5′-TCC CCT CAC CCT CCC AGT AT-3′); ALP (Forward: 5′-ATG GGA TGG GTG TCT CCA CA-3′, Reverse: 5′-CCA CGA AGG GGA ACT TGT C-3′); Runx2 (Forward: 5′-TCC TGT AGA TCC GAG CAC CA-3′, Reverse: 5′-CTG CTG CTG TTG TTG CTG TT-3′); OCN (Forward: 5′-CTC TGC CTT AAA CAC ACA TTG-3′, Reverse: 5′-TTC CCT TTG CCC ACC TC-3′); VEGF (Forward: 5′-ATG AAC TTT GCT GCC AGC TTG-3′, Reverse: 5′-AAT TGT GTT GCG TCA CAT GC-3′); GAPDH (Forward: 5′-ATG AGA AGT ATG ACA ACA GCC-3′, Reverse: 5′-AGT CCT TCC ACG ATA CCA AA-3′).

### 2.8. Western Blotting

PDLSCs were treated with 500 μg/mL DAE for seven days, and total cell protein was extracted by lysing the cells in radioimmunoprecipitation assay buffer (RIPA buffer). Protein concentrations were determined using Bradford protein assays with spectrophotometry at an absorbance of 595 nm. Samples (20 mg total protein) were subjected to SDS–polyacrylamide gel electrophoresis (SDS-PAGE) in a 10% gel, electrically transferred onto polyvinylidene fluoride (PVDF) membranes (Millipore, Billerica, MA, USA) using 80 volts for 100 min, blocked with blocking buffer (5% skimmed milk powder in tris-buffered saline [TBS] containing 0.1% [*v/v*] Tween-20) for 1 h, and then probed with the primary antibody (OCN, 1:500; Abcam, Cambridge, UK) overnight at 4 °C. Blots were washed and placed on an orbital shaker for 1 h with secondary antibodies (IRDye800-conjugated, 1:10,000; Abcam), and then washed with 0.1% Tween-20 in TBS. The proteins were visualized via chemiluminescence (LAS-4000; Fujifilm, Minato, Japan) using the Amersham ECL Plus Western Blotting Detection kit (GE Healthcare, Chicago, IL, USA) according to the manufacturer’s instructions. β-actin was used as a loading control for protein expression in the treated cells.

### 2.9. Statistical Analysis

All experiments were performed thrice for different samples. All data are presented as means ± standard deviation (SD) measured through ANOVA. Statistical comparisons were performed, and P values smaller than 0.05 were considered significant.

## 3. Results

### 3.1. Characterization of PDLSCs

PDLSCs were isolated from enzyme-disaggregated periodontal ligaments of third molars obtained from seven-year-old children. The subcultured cells, that is, the cells obtained from a single colony after isolation, were gradually cultivated as an adherent monolayer and exhibited a fibroblast-like morphology. These PDLSCs were positive for expression of the important MSCs markers CD166 (99.02%), CD90 (99.50%), and CD73 (98.95%), early mesenchymal stem cell marker CD146 (76.64%), and for the endothelial progenitor marker CD105 (94.89%); however, they did not express the hematopoietic markers CD34 (0%) or CD45 (1.08%). Based on these results, the isolated PDLSCs cells are consistent with the cells purified by Gay et al. which could differentiate into multiple cell lineages, similar to BMSCs [20].

### 3.2. Optimization of Microwave-Assisted DAE Extraction

Using 5 g of ground sample, the parameters of the MAE system were modified to determine the optimal extraction procedure. Figure 1 indicates the different yields of DAE at different solid/solvent ratios, operative temperatures, extraction times, and microwave power levels. Upon final evaluation, the optimal extraction conditions were determined to be 500 W, 30 min, 50 °C, and a 1:15 solid/liquid ratio; the maximal extracted yields were 58.66 ± 0.22% (*w/w*).

### 3.3. DAE Induces Osteogenic Differentiation

Cultured PDLSCs were treated with different concentrations of DAE, and cell metabolic activity was measured by MTT on days one, four, and seven (Figure 2a). PDLSCs could proliferate stably in different concentrations of DAE treatment, indicating that DAE has no major side effects on cellular proliferation. However, the results also show that as the DAE concentration rises, the toxicity to the cells becomes stronger (Figure 2b). Considering the long-term culture, 500 µg/mL would be used as the treatment dose for subsequent experiments based on the 7-day cell survival rate of 92%. After induction with 500 µg/mL DAE, changes in cell morphology were observed on days 7, 14, and 21 (Figure 2c). The DAE-induced cells tended to be spindle-shaped, whereas the control cells became more and more slender.

### 3.4. Osteoblastic Gene Expression

Differences in expression of various osteogenesis-related genes from day 0 to 21 were revealed using RT-PCR (Figure 3; *p* < 0.05). The levels of gene expression were quantified as fold-changes in expression using the housekeeping gene GAPDH as a control. The expression levels of Col-1, Runx2, and ALP all reached a maximum at seven days, then gradually decreased with time. These results indicate that DAE can rapidly induce PDLSCs to undergo osteogenic differentiation. Simultaneously, DAE was also found to promote vascular endothelial growth factor (VEGF), which likewise reached its peak at seven days and then decreased. This indicates that the osteogenic differentiation induced by DAE may be related to VEGF.

### 3.5. Calcium Quantification by OCN Staining

Alizarin Red S is used to stain intracellular Ca as well as Ca-binding proteins and proteoglycans, and it is useful in evaluating osteoblastic differentiation. The purple dots shown in Figure 4a reveal the mineralized deposits, which represent mature bone cells and are indicators of late osteoblastic differentiation. The quantified results after cetylpyridinium chloride dissolution and staining are shown in Figure 4b (*p* < 0.05). The calcification deposition in the control group was found not to increase after 14 days, while in the DAE-induced group this continued to increase, and the mineralization deposition was higher than that in the control group from day seven onwards.

OCN has the greatest osteogenic effects among all the markers related to osteogenic differentiation. Only mature bone cells can express a large number of OCN genes and then deposit mature bone matrix. Showing the results of qRT-PCR, Figure 4c (*p* < 0.05) indicates that the expression of OCN genes in the DAE-induced group was 1.6-fold greater than the control group after 21 days of culture. Similarly, it can be clearly observed from Figure 4d,e (*p* < 0.05) that the protein expression of OCN is also elevated compared to the control group, as shown by the western blot assay.

### 3.6. PI3K Pathway Assay

RT-PCR was performed to quantify the expression of pre-osteoblastic genes in cells cultured with DAE for three days after pretreatment with PI3K inhibitor (LY294002) for 1 h (Figure 5). Following inhibitor treatment, DAE did not induce Col-1, ALP, and Runx2 markers of osteogenesis, proving that PI3K is an important component of DAE-induced osteogenesis.

### 3.7. Optimization of Electrospinning

In order to form the Taylor cone for stable spinning, various solute and solvent ratios were assessed. Figure 6a shows that for the optimal PCL/PEO nanofiber with 522 ± 159 nm diameter; the most appropriate conditions were a solute ratio PCL:PEO = 8:2, a solvent ratio CF:DMF = 8:2, and 16 kV. Figure 6b shows that for the optimal PCL/PEO/DAE nanofiber with 488 ± 197 nm diameter; the most appropriate conditions were a solute ratio PCL:PEO:DAE = 8:2:1, a solvent ratio CF:DMF:DMSO = 8:1:1, and 12 kV. The addition of DAE did not significantly change the diameter of nanofibers. The structure of the degraded fibers under SEM is shown in Figure 6c,d. PCL/PEO maintains an original appearance of fiber filament, and only a small amount of the surface layer is degraded; meanwhile, the PCL/PEO/DAE has a relatively degraded structure, with some broken fiber filaments.

### 3.8. Characterization of Nanofibers

FTIR was used to detect the functional groups of the three samples: DAE, PCL/PEO, and PCL/PEO/DAE (Figure 7a). Both PCL/PEO and PCL/PEO/DAE nanofibers showed the absorption peaks of C−O at 1100 cm^−1^ and C=O at 1735 cm^−1^; these values are characteristic peaks of PCL and PEO. The index component of DAE, akebia saponin, contains a large number of O−H groups; the characteristic peaks of DAE appeared at 3100–3700 cm^−1^ due to molecular stretching vibrations. It was confirmed that the DAE was successfully mixed and existed in the nanofibers.

The results of the swelling assay are shown in Figure 7b. PCL/PEO has a swelling degree of 5.06, which means that a unit weight of fiber membrane can absorb 5.06-fold that weight of water. However, after the addition of DAE, the degree of swelling rose further to 5.61. After adding DAE, the porosity of the fiber membrane also increased slightly (Figure 7c), indicating that in addition to the hydrophilicity of DAE itself, this addition could also increase the porosity, resulting in an increased degree of swelling. A tensile assay was performed on the nanofibers with the same test piece specifications, and the mechanical properties with and without the addition of DAE were compared. Figure 7d clearly indicates that the tensile strength of PCL/PEO/DAE is significantly lower, while in contrast, the elongation is significantly increased. The Young’s modulus of the PCL/PEO nanofibers was 1.4-fold greater than that of the PCL/PEO/DAE nanofibers. Figure 7e shows that the degradation trends of the two fibers are roughly the same; both fibers degraded rapidly over the first three days, then gradually slowed down. 

### 3.9. Nanofibers Containing DAE Induce Osteogenic Differentiation

Figure 8a shows that the PDLSCs attached suitably on the PCL/PEO and PCL/PEO/DAE nanofibers after two days in culture. Cells continued to grow and proliferate on the nanofibers, and the cell viability measured by CCK-8 assay is shown in Figure 8b. These results confirm that DAE is not toxic to PDLSCs, and that PCL/PEO/DAE nanofibers are suitable for cell attachment, spread, and proliferation.

PDLSCs induced by DAE on nanofibers also successfully differentiated into osteoblasts. PDLSCs were cultured on two fibrous nanofibers for seven days, and the expression levels of four index genes with and without DAE were determined by qRT-PCR assay. Figure 8c indicates that, although the expression of Col-1, ALP, and Runx2 were higher in the early and middle stages of osteogenic differentiation in PCL/PEO, the expression of OCN is higher in the later stage of osteogenic differentiation in PCL/PEO/DAE, indicating that many of the cells cultured on PCL/PEO/DAE nanofibers had been rapidly induced to the late stage of osteogenesis. Figure 8d shows the mineralized deposits revealed by Alizarin Red S staining, and the calcium quantification is shown in Figure 8e. Similarly, the expression of OCN protein in the western blot assay (Figure 8f) demonstrates that cells cultured on PCL/PEO nanofibers have more OCN expression than cells cultured on a dish without nanofibers, but shows that PCL/PEO/DAE nanofibers further promotes osteogenesis, resulting in an even higher expression of OCN protein. Quantitative results of mineralized deposition staining are shown in Figure 8g; these show that the amount of deposition on PCL/PEO/DAE is 2.6 higher than that on PCL/PEO, and this difference is statistically significant. The results show that PCL/PEO nanofibers can promote osteogenic differentiation, but the effect of PCL/PEO/DAE in inducing stem cell differentiation is more significant.

## 4. Discussion

DA is the root of the perennial herb *Dipsacus apercides* C.Y. Cheng et T.M. Ai [25]. The decoction of DA-containing crude saponin extract has obvious effects on promoting bone healing. DAE is reportedly able to improve local blood circulation, promote the absorption and mechanization of hematomas, promote the proliferation of chondrocytes, accelerate the synthesis of various types of collagen, improve the structure and arrangement of collagen, affect the synthesis of collagen during bone fracture healing in both quantity and quality, and promote fracture healing [26].

Compared with general extraction methods, MAE achieves higher yields, uses less solvents, and consumes less energy. MAE has achieved remarkable results in the areas of chemical synthesis, biomedicine, and manufacturing of new materials. This technique can convert electromagnetic energy into heat at a specific frequency and temperature to rapidly heat the reaction materials. In this study, the optimal parameters for intermittent MAE per 1 g of powder were found to be as follows: 15 mL of 70% ethanol, extracted at 50 °C and 500 W for 30 min. Using these parameters, a 58.66% (*w/w*) crude extraction product yield was obtained.

PDLSCs are a type of human periodontal ligament-derived mesenchymal stem cell. Many studies have used these cells as a tool for bone differentiation research [27,28], and successful cell therapy studies have been conducted in pigs [29] and dogs [30]. Therefore, PDLSCs are extremely suitable for osteogenesis-related research. The dosage of DAE used in experiments was determined by the results of a cell viability assay (MTT assay), which revealed 500 μg/mL as the optimal concentration to be used for subsequent tests. Taken together, the results of the mineralization assay, gene expression analysis, and western blotting assay prove that DAE does induce the differentiation of PDLSCs into mature bone cells. The cellular morphology changes observed in this study are consistent with the study of Rodríguez et al. [31]. As a result of qRT-PCR, DAE was also unexpectedly found to increase the expression of VEGF, indicating that DAE may stimulate osteogenesis through the promotion of VEGF-related pathways [32]. Deckers et al. reported that VEGF may be present in small amounts during the early stage of osteogenesis but does not directly participate in the osteogenic process [33]. In addition, Mayr-Wohlfart et al. found that VEGF can significantly increase the proliferation of osteoblasts [34], and Shibuya indicated that VEGF usually promotes cell proliferation through a PI3K/Akt-related pathway [35]. In our study, after the PI3K inhibitor LY294002 was added, the expression levels of osteogenic genes were lower than those in the untreated group, indicating that PI3K is indeed an essential element for DAE-induced osteogenesis of PDLSCs. Therefore, it can be surmised that DAE induces PDLSCs to stimulate endogenous production of VEGF and activate the PI3K/Akt pathway to regulate downstream eNOS or mTOR, promoting osteogenic differentiation.

Since DA contains saponins that have extremely high polarity and are insoluble in low-polarity CF and DMF, DMF was partially replaced with DMSO; a final optimal solvent ratio of CF:DMF:DMSO = 8:1:1 was determined for PCL/PEO/DAE nanofiber electrospinning. The average fiber diameters of PCL/PEO and PCL/PEO/DAE were 522 nm and 488 nm, respectively. These fiber diameters were even finer than those reported by Remya et al. [6], indicating that CF is more suitable as a solvent for PCL/PEO than DCM. In addition, it has been found that DAE can increase the conductivity of the solution, which is consistent with most studies of extractive addition [36]. In the FTIR analysis, through the comparison of characteristic peaks, DAE was confirmed to be fully mixed into the fibers. In the tensile, degradation, and swelling experiments, it was found that after adding DAE, the ductility, degradability, and swelling were all significantly improved, while the tensile strength was significantly decreased. The trends of mechanical strength and degradability were similar to those reported by Pajoumshariati et al., who modified PCL with the addition of ginseng extract [37]. In our study, the 21-day cell viability assay proved that the added DAE did not increase the toxicity of the fibers. Through the ARS staining assay, the PDLSCs growing on PCL/PEO/DAE showed more mineralized deposits, approximately 2.6-fold more than cells grown on PCL/PEO. Comparing the differences in bone protein and gene expression between the two fibrous membranes and culture plates, we also found that PDLSCs do indeed have an increased osteogenic differentiation tendency when cultured on PCL/PEO fibers. Col-1 decreased in the early stage of osteogenesis, while ALP and Runx2 increased in the intermediate stage and OCN increased in the later stage. However, this tendency was more obvious on PCL/PEO/DAE fibers. These results indicate that nanofiber scaffolds have the effect of promoting osteogenic differentiation of PDLSCs, and that the osteogenic differentiation effect is more significant after adding DAE, indicating that DAE promotes the osteogenic differentiation of PDLSCs. In this study, only 9% (*w/w*) of DAE was added to the nanofiber, which did not optimize this parameter. The amount of DAE can be changed to further promote the osteogenesis effect.

## 5. Conclusions

Greater amounts of DA can be extracted than previously thought, as a 58.66% (*w/w*) DAE yield was obtained under the optimal conditions of MAE. Osteogenic differentiation of PDLSCs treated with 500 μg/mL DAE resulted in a 1.42-fold increase in the number of mineralized deposits and a 1.6-fold increase in the gene expression of OCN. The actions of DAE on PDLSCs may be through the VEGF/PI3K/Akt pathway, activating downstream eNOS or mTOR to promote cell proliferation and bone formation. The addition of DAE effectively increases the elongation, swelling, porosity, and degradability of PCL/PEO nanofibers, resulting in the fibers having better applicability as tissue engineering scaffolds and increased osteogenesis induction effects.

## Figures and Tables

**Figure 1 polymers-13-02245-f001:**
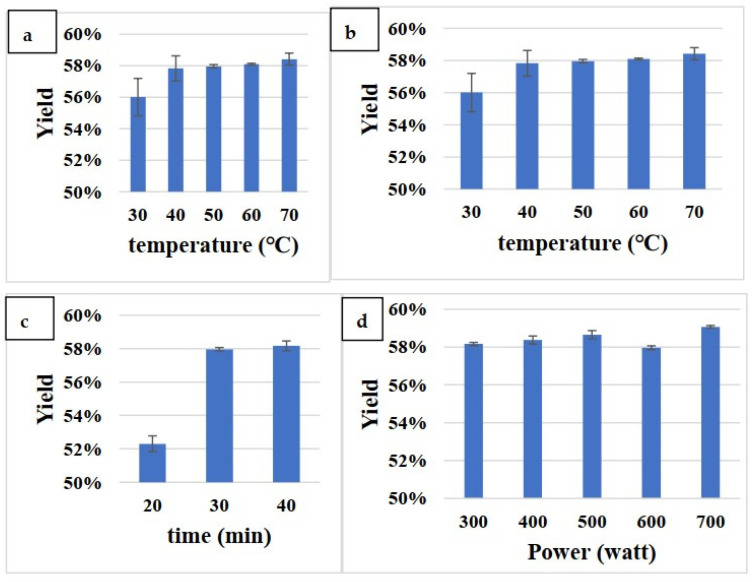
The extract yield of various parameters of MAE: solid/solvent ratio (**a**), operating temperature (**b**), extraction time (**c**), and microwave power (**d**). Values are expressed as mean ± SD (n = 3).

**Figure 2 polymers-13-02245-f002:**
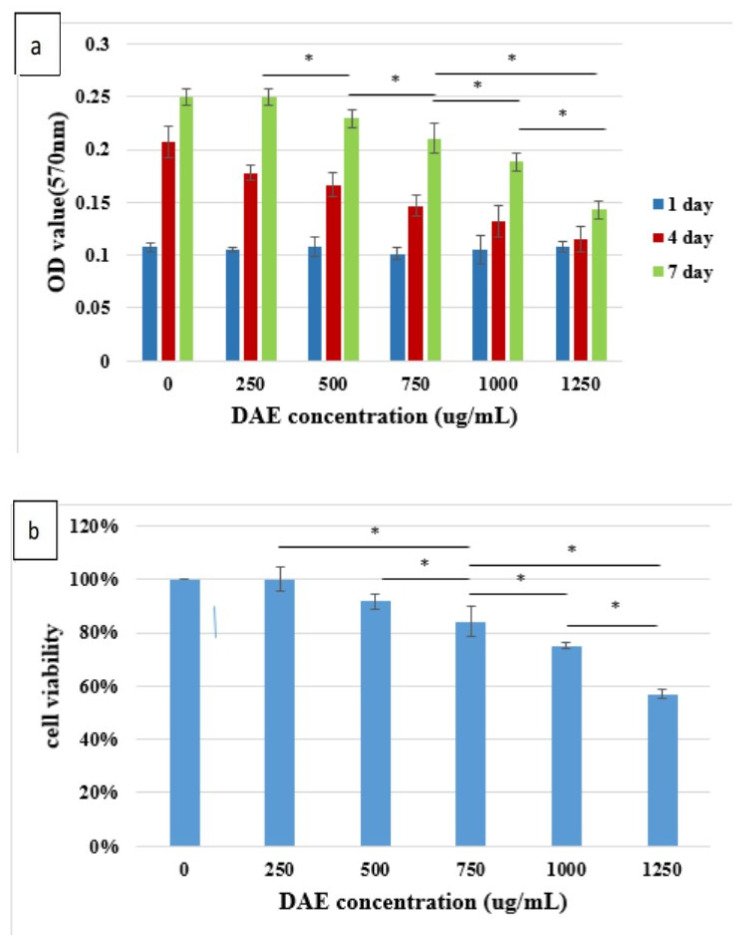
The proliferation of PDLSCs treated with different concentrations of DAE (**a**), and cell viability (**b**), * *p* < 0.05 compared each other. Values are expressed as mean ± SD (n = 3). The cellular morphology changes before and after treatment with 500 µg/mL DAE (**c**); scale bar = 100 μm.

**Figure 3 polymers-13-02245-f003:**
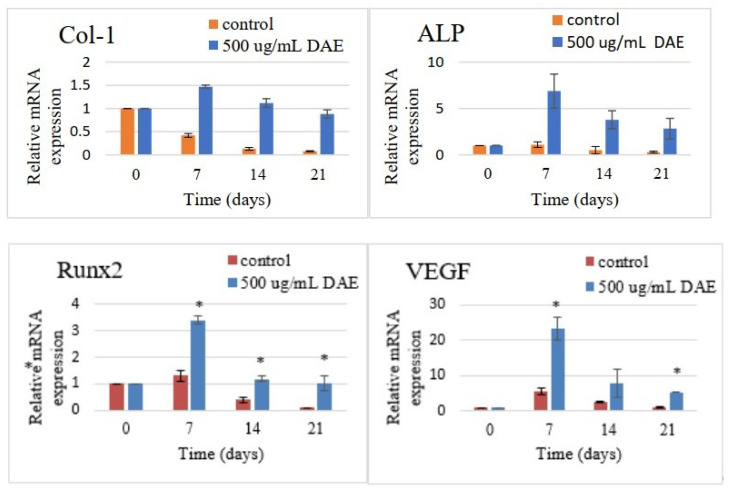
Quantitative RT-PCR analysis of the expression of the osteogenesis-related genes Col-1, ALP, Runx2, and VEGF after differentiation of PDLSCs induced with DAE. Values are expressed as mean ± SD (n = 3), * *p* < 0.05 for DAE group compared with control group.

**Figure 4 polymers-13-02245-f004:**
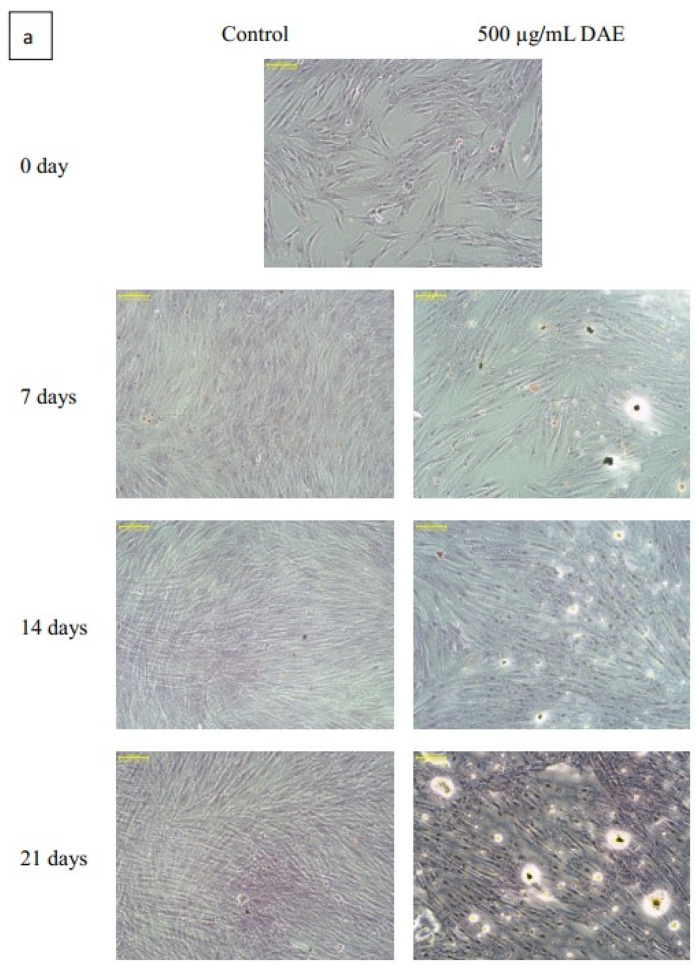
The Alizarin Red S staining (**a**); scale bar = 100 μm and quantification (**b**) of PDLSCs induced with DAE. Quantitative RT-PCR analysis of OCN gene expression (**c**). Western blotting of the OCN protein (**d**) and quantification (**e**). Values are expressed as mean ± SD (n = 3), * *p* < 0.05 for DAE group compared with control group.

**Figure 5 polymers-13-02245-f005:**
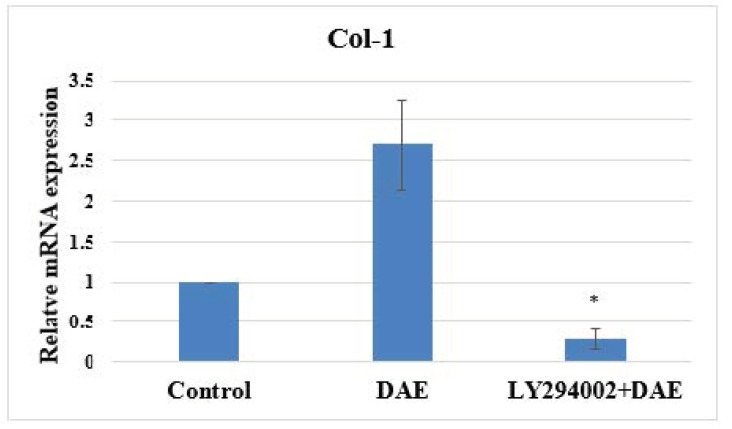
Quantitative RT-PCR analysis of the Col-1, ALP, and Runx2 genes after treatment of PDLSCs with PI3K inhibitor. Values are expressed as mean ± SD (n = 3), * *p* < 0.05 for compared with control group.

**Figure 6 polymers-13-02245-f006:**
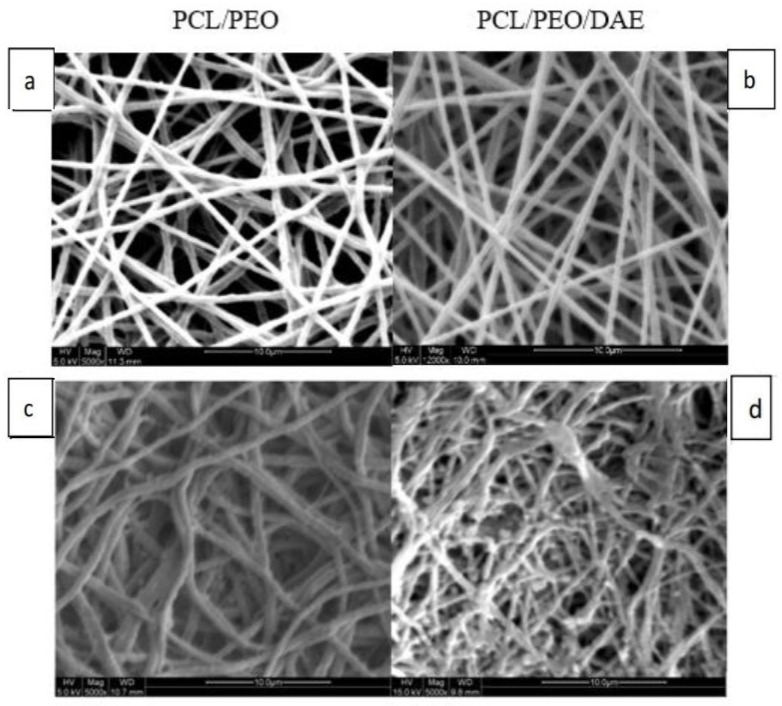
SEM micrograph of PCL/PEO (**a**) and PCL/PEO/DAE nanofibers (**b**); degraded PCL/PEO (**c**) and degraded PCL/PEO/DAE (**d**), scale bar = 10 μm.

**Figure 7 polymers-13-02245-f007:**
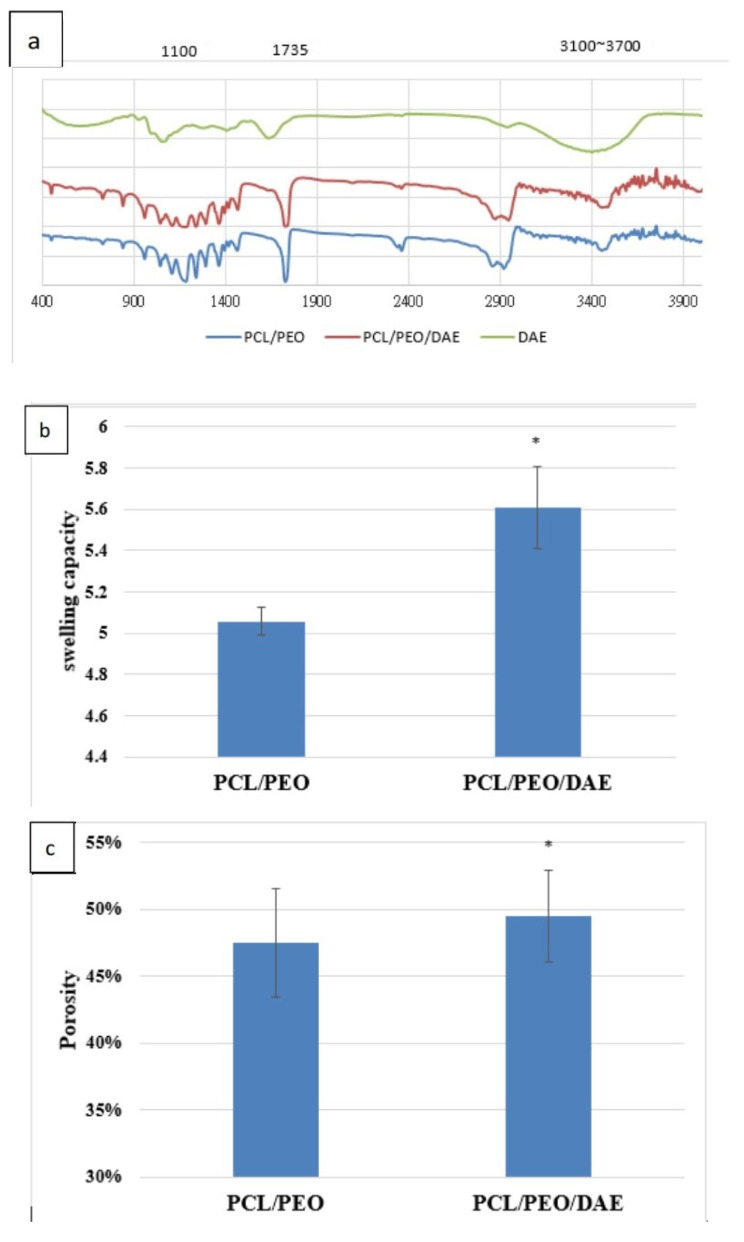
FTIR analysis of various nanofibers and DAE (**a**), the swelling ratio (**b**), porosity (**c**), mechanical property (**d**), and biodegradation rate (**e**) of the two nanofibers. Values are expressed as mean ± SD (n = 3), * *p* < 0.05 compared with PCL/PEO group.

**Figure 8 polymers-13-02245-f008:**
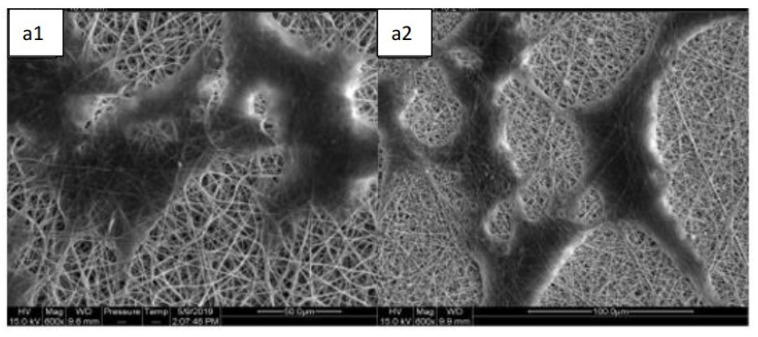
PDLSCs grown on the PCL/PEO (**a**1) and PCL/PEO/DAE nanofiber (**a**2) over two days of culture; scale bar = 100 μm. Cell viability (**b**) * *p* < 0.05 for compared with culture day 1. Quantitative RT-PCR analysis of osteogenesis-related genes (**c**), Alizarin Red S staining of PDLSCs cultured on the PCL/PEO (**d1**) and PCL/PEO/DAE nanofiber (**d2**) over 21 days: scale bar = 100 μm. The quantification (**e**), and western blotting (**f**) and quantification (**g**) of PDLSCs cultured on the nanofibers treated with 500 μg/mL DAE. Values are expressed as mean ± SD (n = 3), * *p* < 0.05 compared with control.

## Data Availability

The data presented in this study are available on request from the corresponding author.

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
