# Peer review of "Polycaprolactone/Polyethylene Glycol Blended with Dipsacus asper Wall Extract Nanofibers Promote Osteogenic Differentiation of Periodontal Ligament Stem Cells"

_polymers, 2021, doi:10.3390/polym13142245_

Round 1

Reviewer 1 Report

In the manuscript “Polycaprolactone/polyethylene glycol blended with Dipsacus asper wall extract nanofibers promote osteogenic differentiation of periodontal ligament stem cells” submitted to “Polymers” for potential publication, authors have investigated the effect of Dipsacus extract for promoting osteoblastic differentiation of periodontal ligament stem cells. This is a well designed study and well presented manuscript.

The study requires some further improvements in the presentation:

Abstract require changes: the objective is not clear and presented a lot of abbreviations; please add  a clear objective of the study and double check that all the abbreviations are defined at their first appearance, also try to reduce number of abbreviation if possible.

Introduction is missing to support the objective and rationale of the study. The authors should explain more clearly the rationale for using PDLSCs and not any other stem cells in this study.

-Please carefully check the use of abbreviations in the main text also and figures as the figures did not mention the abbreviations used.

-A paragraph should be added on the role of nanofibers and regeneration treatment options in periodontal defects mainly in the third paragraph, following papers are beneficial and can be included to improve/include this information  :

Long-term Clinical Performance of Regeneration versus Conservative Surgery in the Treatment of Infra-bony Defects: A Systematic Review. J. Int. Acad. Periodontol, 23, pp.31-56.

Clinical effectiveness of anorganic bovine-derived hydroxyapatite matrix/cell-binding peptide grafts for regeneration of periodontal defects: a systematic review and meta-analysis. Regenerative Medicine, 15(2020). DOI: https://doi.org/10.2217/rme-2020-0113.

Regenerative Potential of Enamel Matrix Protein Derivative and Acellular Dermal Matrix for Gingival Recession: A Systematic Review and Meta-Analysis. Proteomes, 9(1), p.11.

Potential of electrospun nanofibers for biomedical and dental applications. Materials, 9(2), p.73.

Methods and results are described very well;

Section 2.3: how much the electric potential was used? During electrospinning?

How the electrospun membranes were sepearated from the target/foil? Such details should be added.

Figure 3 and 4: the scale bars in images are very thin and not readable. Please enlarge/bold them if possible. Figure 4 can be better presented,.

-The authors should include future research directions and limitations of this study.

Author Response

Response to Reviewers

The authors would like to express my sincerely thanks to the editor and reviewers for the professional comments and constructive suggests. The article was revised following the suggestions. The modifications are listed as below.

Reviewer#1:

  1. Abstract require changes: the objective is not clear and presented a lot of abbreviations; please add a clear objective of the study and double check that all the abbreviations are defined at their first appearance, also try to reduce number of abbreviation if possible.

Response: We have added the objective of the study and carefully check the abbreviations.

  1. Introduction is missing to support the objective and rationale of the study. The authors should explain more clearly the rationale for using PDLSCs and not any other stem cells in this study.

-Please carefully check the use of abbreviations in the main text also and figures as the figures did not mention the abbreviations used.

-A paragraph should be added on the role of nanofibers and regeneration treatment options in periodontal defects mainly in the third paragraph, following papers are beneficial and can be included to improve/include this information.

Response: We have deleted unnecessary narratives to highlight the objective and rationale. In additional, we have carefully check the use of abbreviations in the main text and figures, and have added the statement of nanofiber in the third paragraph of Introduction.

  1. Section 2.3: how much the electric potential was used? During electrospinning? How the electrospun membranes were sepearated from the target/foil? Such details should be added.

Response: The solute ration, solvent ration and electric voltage will be adjusted to the best condition of electrospinning, and the results were shown in the Optimization of electrospinning section of Result. The membrane is removed from the aluminum foil and this statement has added to manuscript.

  1. Figure 3 and 4: the scale bars in images are very thin and not readable. Please enlarge/bold them if possible. Figure 4 can be better presented,

Response: We have revised the scale in Fig 3 and 4.

  1. The authors should include future research directions and limitations of this study.

Response: We have added the statement in the Discussion.

Reviewer 2 Report

In this manuscript, microwave-assisted alcohol extraction of DA powder was conducted. After optimizing extraction conditions, 58.7% yield of the crude extract was obtained. The effects of DA extract (DAE) on osteogenic differentiation of PDLSCs were intensively studied, including evaluation of marker genes and proteins expressions. Results showed that DAE promoted osteogenic differentiation of PDLSCs. DAE was further entrapped in PCL/PEO electrospun nanofibers. Properties of nanofibers were characterized using FTIR, mechanical strength, biodegradability, swelling ratio and porosity, and cell compatibility. It was found that nanofibers with DAE also promoted osteogenic differentiation of PDLSCs. There were many results reported in the manuscript. However, neither the figures nor texts were well organized.

  1. The objective of this research was not well written in Introduction part. For example, are there any advantages for microwave-assisted alcohol extraction method? What are the possible applications for DAE loaded PCL/PEO electrospun nanofibers?

  1. The introduction and corresponding characterizations for DAE were inadequate. Ie, is DAE a mixture? What is the main component in DAE?

  1. There were some unclear experiment descriptions. How to quantify the amount of DAE when referring to the yield? Molecular weights for PCL and PEO were not included. It was not clear how to conduct the swelling test and how to induce osteogenic differentiation of PDLSCs was not mentioned either.

Author Response

Response to Reviewers

The authors would like to express my sincerely thanks to the editor and reviewers for the professional comments and constructive suggests. The article was revised following the suggestions. The modifications are listed as below.

Reviewer#2:

  1. The objective of this research was not well written in Introduction part. For example, are there any advantages for microwave-assisted alcohol extraction method? What are the possible applications for DAE loaded PCL/PEO electrospun nanofibers?

Response: We have added the statement of microwave-assisted extraction and PCL/PEO electrospun nanofibers application in the revised manuscript.

  1. The introduction and corresponding characterizations for DAE were inadequate. Ie, is DAE a mixture? What is the main component in DAE?

Response: We have added the statement of DAE characterizations, "Alcohol extraction of DA can get a mixture, which mainly contains polysaccharides and triterpene saponins" in the revised manuscript.

  1. There were some unclear experiment descriptions. How to quantify the amount of DAE when referring to the yield? Molecular weights for PCL and PEO were not included. It was not clear how to conduct the swelling test and how to induce osteogenic differentiation of PDLSCs was not mentioned either.

    Response: The yield is extract weight per sample weight, so the yield in manuscript is revised to " 558.66 ± 0.22% (w/w) ". Mw of PCL is 80000 g/mol; PEO is 300000 g/mol. The swelling test was stated in the section "Characterization of nanofiber of Materials and Method" and the induce osteogenic differentiation of PDLSCs was only by the Dipsacus asper wall extract (DAE).

Reviewer 3 Report

The manuscript “Polycaprolactone/polyethylene glycol blended with Dipsacus asper wall extract nanofibers promote osteogenic differentiation of periodontal ligament stem cells” by

Te-Yang Huang et al. describes microwave-assisted alcohol extraction of dipsacus asper blend with PCL and PEO as an excellent source for osteogenesis and mineralization. The authors presented a physical and biological characterization of these fibers including FTIR, mechanical strength, degradation, swelling, and biocompatibility.

  • The abstract was written very poorly and doesn’t properly reflect a short version of the study: why this topic was chosen; how it was conducted; significance of findings; and what will be the next step.

  • The introduction body was written so disperse and disconnected. The authors started with DA and how it works on osteogenesis molecular pathway, and then they switch to PCL, electrospinning; then they switch topic to stem cells, their potentials and osteogenesis differentiation. Overall, the paragraphs are so disconnected from each other.

  • Figures: Overall the figures were prepared in low quality. I highly recommend authors to look at prior publications from MDPI journal and their figures structures and formatting.

  • In Figure 2, to show the significance of findings on proliferation of PDLSCs treated with different concentrations of DAE, it is important to do statistical analysis among different groups (describing the cause of toxicity for concentrations above >750 ug/mL.

  • For Figure 2C, the cellular morphological images are showing cells at confluency state. To show a more detailed image of their phenotypic behavior, I suggest authors to look at earlier time point (day 4) as they presented in their graph. Additionally, the figure needs to be reformatted more professionally.

  • In Figure 3. Quantitative RT-PCR analysis of osteogenesis-related genes. The authors described the statistical analysis description (* p < 0.05) for their comparisons however no comparisons were found among displayed groups.

  • For figure 4, images need to be reformatted and statistical analysis for each figure (4.b, c, e) needs to be included and properly described in the text.

  • For figure 5. Quantitative RT-PCR osteogenesis related genes, statistical analysis for each comparison needs to be calculated, displayed on the graph and described in the text.

  • Figure 6 is so crowded. It needs to be splitted into two figures and properly formatted.

  • Additionally, as part of nanofibers characterization, it is important to check whether the diameter of the fibers change as DAE get incorporated into the formulation. Statistical analysis for Figure 6.d and 6.e is also missing.

  • Statistical analysis is also missing for Figure 7. Extensive analysis is required to further explain the significance of findings.

  • Provided Conclusion for the manuscript needs to be expanded to cover future direction of this research and how these findings can help the field.

Author Response

Response to Reviewers

The authors would like to express my sincerely thanks to the editor and reviewers for the professional comments and constructive suggests. The article was revised following the suggestions. The modifications are listed as below.

Reviewer#3:

  1. The abstract was written very poorly and doesn’t properly reflect a short version of the study: why this topic was chosen; how it was conducted; significance of findings; and what will be the next step.

Response: We have made appropriate revisions to the abstract based on the review’s recommendations.

  1. The introduction body was written so disperse and disconnected. The authors started with DA and how it works on osteogenesis molecular pathway, and then they switch to PCL, electrospinning; then they switch topic to stem cells, their potentials and osteogenesis differentiation. Overall, the paragraphs are so disconnected from each other.

Response: According to the reviewers’ doubts, we have made appropriate revision to the introduction to make the connection of each paragraphs.

  1. Figures: Overall the figures were prepared in low quality. I highly recommend authors to look at prior publications from MDPI journal and their figures structures and formatting.

Response: We have revised the figure quality, based reviewer’ suggestion.

  1. In Figure 2, to show the significance of findings on proliferation of PDLSCs treated with different concentrations of DAE, it is important to do statistical analysis among different groups (describing the cause of toxicity for concentrations above >750 ug/mL.

Response: According to the reviewer’s suggestions, we have re-doed the statistical analysis between different groups in Fig. 2.

  1. For Figure 2C, the cellular morphological images are showing cells at confluency state. To show a more detailed image of their phenotypic behavior, I suggest authors to look at earlier time point (day 4) as they presented in their graph. Additionally, the figure needs to be reformatted more professionally.

Response: Because bone differentiation induction requires a long period of culture, no photos were taken on the 4th day. We try to enlarge the picture so that readers can easily distinguish the changes in cell morphology.

  1. In Figure 3. Quantitative RT-PCR analysis of osteogenesis-related genes. The authors described the statistical analysis description (* p < 0.05) for their comparisons however no comparisons were found among displayed groups.

Response: Due to an error in the conversion process of word to PDF, it has been corrected.

  1. For figure 4, images need to be reformatted and statistical analysis for each figure (4.b, c, e) needs to be included and properly described in the text.

Response: The figure 4a has revised, and because an error in the conversion process of word to PDF, it has been corrected for Fig 4 b,c,e and described in text.

  1. For figure 5. Quantitative RT-PCR osteogenesis related genes, statistical analysis for each comparison needs to be calculated, displayed on the graph and described in the text.

Response: Due to an error in the conversion process of word to PDF, it has been corrected. We also describe in the text.

  1. Figure 6 is so crowded. It needs to be splitted into two figures and properly formatted.

Response: We have divided Figure 6 into Figure 6 and Figure 7, according to the reviewer’s suggestions.

  1. Additionally, as part of nanofibers characterization, it is important to check whether the diameter of the fibers change as DAE get incorporated into the formulation. Statistical analysis for Figure 6.d and 6.e is also missing.

Response: The addition of DAE did not significantly change the diameter of nanofibers. Statistical analysis is an error in the conversion process of word to PDF, it has been corrected.

  1. Statistical analysis is also missing for Figure 7. Extensive analysis is required to further explain the significance of findings.

Response: Due to an error in the conversion process of word to PDF, it has been corrected. We also explain the significance of findings in the manuscript.

  1. Provided Conclusion for the manuscript needs to be expanded to cover future direction of this research and how these findings can help the field.

Response: We have added the future direction of research in the discussion.

Round 2

Reviewer 1 Report

Many thanks for the revision and incorporating all suggested changes to the manuscript

Author Response

Reviewer#1:

  1. Many thanks for the revision and incorporating all suggested changes to the manuscript.

Response: We are very grateful to the reviewer for your professional evaluation and suggestions, and affirmed the revised content. In addition, we also carefully reviewed and corrected the English spelling in the article.

Reviewer 2 Report

All issues have been addressed.

Author Response

Reviewer#2:

  1. All issues have been addressed.

Response: We are very grateful to the reviewer for your professional evaluation and suggestions, and affirmed the revised content. In addition, we also carefully reviewed and corrected the English spelling in the article.